# Getting MoRE out of Mixture of Language Model Reasoning Experts

**Chenglei Si**[1,4]     **Weijia Shi**[2]     **Chen Zhao**[3]
**Luke Zettlemoyer**[2]     **Jordan Boyd-Graber**[1]
[1] University of Maryland     [2] University of Washington
[3] NYU Shanghai     [4] Stanford University
clsi@stanford.edu

## Abstract

While recent large language models (LLMs) improve on various question answering (QA) datasets, it remains difficult for a single model to generalize across question types that require distinct reasoning abilities. We provide empirical evidence that state-of-the-art LLMs suffer from poor generalizability on reasoning types beyond those seen in the prompt. To remedy this, we propose a Mixture-of-Reasoning-Experts (MoRE) framework that ensembles diverse specialized language models. We specialize the backbone language model with prompts optimized for different reasoning categories, including factual, multihop, mathematical, and commonsense reasoning. Our key insight is to leverage agreement among the specialized experts to select the best answer for each question, or to abstain from answering. This gives MoRE higher accuracy than any single specialized model on a collection of 12 QA datasets from four reasoning types. Beyond generalizability, the interpretable design of MoRE improves selective question answering results compared to baselines without incorporating inter-expert agreement. This framework is also more interpretable and useful to human consumers of QA outputs. Our human study confirms that presenting expert predictions and the answer selection process helps annotators more accurately calibrate when to trust the system's output. We release all code and data to facilitate future work. [1]

## 1 Introduction

Question answering (QA) is one of the most common interactions between humans and AI with a wide range of applications (Gardner et al., 2019). When a QA system is deployed in-the-wild—where users can ask any question—the principal challenges are to handle the diversity of question types while ensuring reliability by only providing answers when the system has a high probability of

---

[1] https://github.com/NoviScl/MoRE

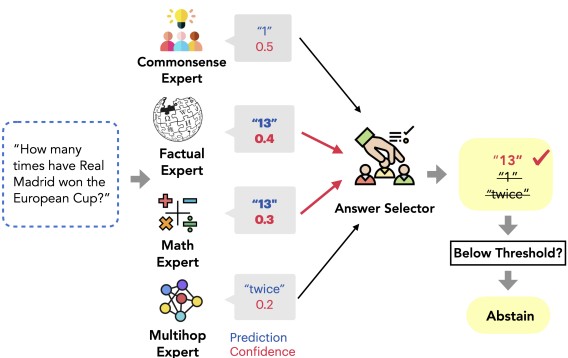

Figure 1: **Overview of MoRE**. In our MoRE framework, each of the four specialized expert models produces a prediction for the test question and we train a classifier to select the best answer among them. The answer selector considers all the *predictions* and their *confidence*, as well as their *agreement* (*e.g.*, in this example the factual expert makes the same prediction as the math expert and so this prediction gets a higher score). Finally, if the selected answer's score is relatively low, the system abstains from answering. In this example, the correct answer should be 14, and MoRE abstained correctly to avoid producing the wrong answer 13.

being correct. This motivates us to develop a QA system that achieves both goals: (1) it should be **generalizable**, adept at handling any type of question; (2) it should answer **selectively**, abstaining from producing erroneous answers.

Toward these goals, one popular approach is to build a unified QA system. While general-purpose LLMs like GPT-3 (OpenAI, 2022) demonstrate impressive question-answering abilities, they lack specialization on particular domains or reasoning types and often fall behind specialized models (Qin et al., 2023; Koco'n et al., 2023). Moreover, to the public, these LLMs are massive black boxes: users have cannot connect the prediction process to whether the outputs are trustworthy.

Therefore, we go against this trend of building a single generalist language model, but rather design a more interpretable system that consists of

a pool of specialized models and each question is answered by one of them. Crucially, to best use complementary strengths of multiple QA models, we implement a pool of diverse and capable specialized models (*e.g.*, by equipping LLMs with corresponding prompting strategies) for each specific reasoning type; then we train a classifier to select the best candidate answer from the specialized models for each question or to abstain from answering (Figure 1). This framework, Mixture-of-Reasoning-Experts (MORE), aims to both generalize and answer selectively.

To obtain the most capable specialist models for each reasoning type, we leverage specialized prompting strategies such as Chain-of-Thought (Wei et al., 2022b) prompting and retrieval-augmented prompting. Experiments on our collection of 12 QA datasets across four diverse reasoning types confirm that our specialist models outperform the backbone model without specialization, but they achieve much lower accuracy on question types outside of their expertise.

With these specialized models, we propose our MORE framework to combine their strengths. MORE selects the best candidate answer from the pool of specialized models, and we teach MORE to abstain from answering if none of the candidate answers are correct. We design our answer selector based on these indicative features: (1) the match between the question type and each specialized model's expertise; (2) the confidence of each specialized model and the characteristics of their predictions; and (3) the agreement among all specialized models, which is a novel feature that we propose. Experiments validate that by ensembling the specialized experts this way, MORE significantly outperforms any single specialized model across all four diverse reasoning types.

Apart from the improved generalizability of MORE, an important byproduct of cross-checking among specialized experts is to offer a useful signal for understanding the whole system's working mechanism. This is validated by the experimental results showing that incorporating agreement among different specialized experts leads to better selective QA results—where the system answers as many questions as possible while maintaining high accuracy—and presenting such internal decision processes to human annotators helps them determine the correctness of the system predictions more accurately and in a shorter time.

## 2 Problem Setup

Given our goal of developing a QA system that generalizes across reasoning types and abstains appropriately, we introduce our task and evaluation details.

### 2.1 Generalizablility Across Reasoning Types

We aim to develop a QA system that handles any type of question with different reasoning challenges. Therefore, we evaluate QA systems from the following representative reasoning categories:

- **Factual reasoning**: factoid questions that are knowledge-intensive.
- **Multihop reasoning**: decomposing the question into sub-steps and reasoning across them.
- **Mathematical reasoning**: mathematical and logical computations, such as math word problems.
- **Commonsense reasoning**: commonsense knowledge that is often implicit.

Our list of QA reasoning types is selected based on existing QA taxonomy (Rogers et al., 2021). The list is not exhaustive – we focus on them partly due to the availability of evaluation benchmarks but our system can be easily extended to other reasoning types. Our final reported metric is based on the macro-average across 12 different datasets from these reasoning types.

### 2.2 Selective Prediction

To deploy the QA system in real-world applications, the system should abstain from answering when its final answer is likely to be wrong. Therefore, we adopt the ***selective*** QA setup (El-Yaniv and Wiener, 2010; Kamath et al., 2020) as our final evaluation setting.[2] More formally, given a question $x$, the QA system returns a predicted answer $\hat{y}$. We assign a score $c \in \mathbb{R}$ to this prediction that reflects the likelihood of this answer being correct. We evaluate selective QA by ranking all predictions by their scores $c$ and abstain if the score $c$ is lower than a threshold $\gamma$. Intuitively, lowering the threshold $\gamma$ would increase the answering coverage, but also incur higher error rates. We introduce metrics for evaluating such trade-offs in Section 5.1.

The crux of the problem is to develop calibrators that can reliably score the predictions to reflect their

---

[2]While our primary focus for evaluation lies in selective QA, in section 4, we also directly compare predicted answers and gold labels as a sanity check without selective prediction.

probability of being correct. This is where the interpretable design of our proposed MORE system helps: we will demonstrate in Section 5 that the inter-expert agreement information in the MORE system is an effective signal for predicting the correctness of answers for both automatic abstention and human verification of answer correctness.

## 3 Mixture of Reasoning Experts

This section introduces our Mixture of Reasoning Experts (MORE) framework, including how to obtain diverse reasoning experts, how to ensemble them, and how to predict answer correctness.

### 3.1 Specialized Reasoning Experts

The first step of our MORE system is to obtain a diverse set of specialized models so that we can combine their strengths via strategic ensembling. Although there are numerous ways of building specialized QA models, we design specialized reasoning experts via prompting a LLM since it has state-of-the-art accuracy on many reasoning tasks. We specialize the Codex model (Chen et al., 2021) for different reasoning types with four specialized prompting methods (the example prompts are listed in the Appendix, Figure 3):

- **Factual expert** with retrieval-augmented prompting. Following Si et al. (2023a), for each question, we retrieve the top 10 most relevant passages from Wikipedia with Contriever (Izacard et al., 2022) and append them to the prompt right before the question.
- **Multihop expert** with Chain-of-Thought (CoT) prompting (Wei et al., 2022b). We add manually-written rationales after each demo question in the prompt to elicit multi-step reasoning process for the questions.
- **Math expert** with CoT prompting. We add the accompanied explanations provided in GSM8K after each demo question in the prompt to elicit similar reasoning steps for the questions.
- **Commonsense expert** with generated knowledge prompting (Liu et al., 2021). We generate 10 fact pieces related to each question using the Codex model and append them to the prompt right before the question.

After obtaining predictions from each expert, we train a classifier to pick the best answer. This allows MORE to ensemble these four specialized expert models without knowing *a priori* the question's reasoning type.

### 3.2 Ensembling via Answer Selection

We combine the strengths of the specialized experts by employing a feature-based random forest classifier to score each candidate answer, the score is used for selecting the final answer and determining when to abstain. We assume the setting where we obtain the predictions from each specialized model first and then select the best answer. We describe the details of training the classifier in this section.

**Feature Set**   We use hand-designed features including the expert type, question characteristics (*e.g.*, the question word, length, and existence of numerical values), answer characteristics (*e.g.*, confidence, length, and the token overlap with questions, contexts, and rationales), and inter-expert agreement. We include the full list of features in the Appendix (Section A.3). Here we highlight the inter-expert agreement features that are uniquely introduced in this work thanks to the more interpretable design of MORE, which includes the frequency of the predicted answer among all four experts' predictions, and the token overlap among these expert predictions.

Additionally, we experiment with a setting where we route the question to the best expert based only on the question itself without obtaining predictions from all experts. In that setting, we train the random forest classifier without using any answer characteristic or inter-expert answer agreement features (more details in Section 4.4).

**Training Data and Objective**   We hold out 100 examples per QA dataset as the training data (1200 examples in total). During training, we extract the features from the questions and the expert models' outputs to train the random forest classifier with a binary classification objective to predict whether the expert model prediction is correct or not. During inference, for each question, we score all experts' answers with this classifier and select the answer with the highest score as the final answer. If the final selected answer's score is below a searched threshold, we abstain from answering.

Apart from the random forest classifier, we also experimented with other feature-based classifiers and finetuning pretrained language models like BERT (Devlin et al., 2019), but found them to be less effective.

| | Factual | | | Multihop | | | Math | | | Commonsense | | | |
|---|---|---|---|---|---|---|---|---|---|---|---|---|---|
| | NQ | TQA | SQuAD | HQA | BeerQA3+ | MuSiQue | GSM8K | SVAMP | MultiArith | CSQA | CSQA2.0 | QASC | Macro-Average |
| *Single Expert Results (Section 4.2)* | | | | | | | | | | | | | |
| Specific Few-Shot | 37.8 | 70.3 | 20.0 | 27.3 | 31.5 | 10.3 | 19.5 | 66.0 | 41.5 | 75.8 | 64.0 | 67.4 | 44.3 |
| Factual Expert | **42.8** | **72.3** | **30.0** | **37.0** | 27.0 | 12.5 | 11.8 | 53.5 | 32.2 | 46.6 | 62.0 | 33.1 | 38.4 |
| Multihop Expert | 34.8 | 61.3 | 19.0 | 34.3 | **46.3** | **15.5** | 37.5 | 70.5 | 75.9 | 55.2 | 62.5 | 54.1 | 47.2 |
| Math Expert | 21.0 | 59.8 | 13.8 | 22.5 | 34.0 | 7.5 | **61.8** | **74.5** | **92.2** | 51.1 | 58.0 | 57.9 | 46.2 |
| Commonsense Expert | 32.5 | 64.0 | 16.3 | 31.3 | 38.5 | 10.8 | 41.5 | 72.5 | 75.4 | **78.4** | **65.3** | **68.9** | 49.6 |
| *Ensemble: Full MORE Router (Section 4.3)* | | | | | | | | | | | | | |
| Oracle | 53.8 | 78.5 | 37.0 | 51.7 | 61.0 | 25.5 | 75.8 | 90.3 | 99.2 | 92.1 | 86.0 | 88.2 | 69.9 |
| Majority Vote | 33.8 | 68.0 | 18.3 | 31.3 | 33.0 | 9.0 | 26.5 | 64.5 | 68.8 | 57.0 | 63.0 | 49.6 | 43.6 |
| MaxProb | 38.8 | 69.3 | 23.3 | 38.5 | 42.5 | 13.5 | 48.5 | 75.3 | 83.9 | 47.6 | 62.0 | 53.4 | 49.7 |
| MORE - Codex Router | 34.5 | 62.7 | 18.5 | 36.0 | 45.3 | 15.3 | 53.8 | 77.0 | 88.7 | 60.8 | 63.0 | 60.7 | 51.4 |
| MORE - RF Router | 39.0 | 71.8 | 25.8 | 37.5 | 46.0 | 14.0 | 63.5 | 80.5 | 95.0 | 78.9 | 66.8 | 72.9 | **57.6** |
| *Ensemble: Question Only Router (Section 4.4)* | | | | | | | | | | | | | |
| Random Selector Baseline | 32.3 | 64.8 | 21.5 | 33.3 | 37.3 | 10.5 | 35.8 | 67.8 | 70.6 | 54.2 | 62.0 | 53.6 | 45.3 |
| Q-Type Oracle | 42.8 | 72.3 | 30.0 | 34.3 | 46.3 | 15.5 | 61.8 | 74.5 | 92.2 | 78.4 | 65.3 | 68.9 | 56.8 |
| MORE - RF Router | 34.5 | 62.7 | 20.5 | 31.5 | 39.0 | 10.8 | 52.3 | 74.3 | 89.2 | 67.7 | 63.5 | 56.4 | 50.2 |

Table 1: Per-dataset accuracy (exact match) breakdown on all 12 QA datasets. We highlight the best single-expert result on each dataset in **bold**. Specialized QA models (first block) excel at the corresponding reasoning types and lose generalizability on others. Our proposed MORE system with the random forest answer selector (second block) has the best macro-average accuracy across all datasets (57.6), beating all specialized QA models, although it still lags behind the oracle ensemble (69.9). MORE with the few-shot Codex router performs significantly worse than the full random forest router (51.4). Notably, MORE with the question-only random forest router (last block) can still outperform the single expert baselines but performs much worse than the full MORE router.

**Few-Shot Answer Selection**  While training a random forest answer selector gives better QA accuracy (as we will show in the next section), it requires a moderate amount of training data. We also explore a few-shot alternative, where we directly prompt the Codex model with 14 randomly selected demo examples, each consisting of the question, the predictions of the four specialized models, and the best answer among them. [3] During inference, we append the question and prompt Codex to select the best answer.

## 4 Sanity Check: MORE Improves Generalizability

This section describes our experiments to verify that MORE's ensemble of diverse experts improves generalizability.

### 4.1 Experimental Setup

**Evaluation Datasets**  We evaluate on 12 datasets covering four reasoning types. Specifically, Natural Questions (NQ) (Kwiatkowski et al., 2019), TriviaQA (Joshi et al., 2017), and SQuAD (Rajpurkar et al., 2016) for factual reasoning; HotpotQA (Yang et al., 2018), BeerQA (Qi et al., 2020),[4] and MuSiQue (Trivedi et al., 2021) for multihop reasoning; GSM8K (Cobbe et al., 2021), SVAMP (Patel

et al., 2021), and MultiArith (Roy and Roth, 2015) for mathematical reasoning; CommonsenseQA (CSQA) (Talmor et al., 2019), CSQA2.0 (Talmor et al., 2021), and QASC (Khot et al., 2019) for commonsense reasoning. For each dataset, we randomly sample 400 questions from the test set for our evaluation to control inference costs.

**Demonstration Examples for Specialized Experts**  We use 16 randomly sampled training examples as demonstration examples of each specialized prompt. Specifically, we use examples from Natural Questions as demonstration examples for the factual expert, examples from HotpotQA for the multihop expert, examples GSM8K for the mach expert, and examples from CSQA for the commonsense expert. These demonstration examples are formatted with the corresponding specialized prompting strategies described above. Additionally, we also include a dataset-specific few-shot baseline where we randomly sample and concatenate 16 question-answer pairs from each corresponding dataset being evaluated as the prompt without any specialized prompting techniques. We use the answer exact match (EM) as the evaluation metric for all datasets.

### 4.2 Specialization and Loss of Generalizability

We first evaluate each of the four specialized reasoning experts on the collection of 12 datasets (Table 1, first block).

---

[3]We only select examples where the correct answer is among the expert predictions.

[4]The original BeerQA dataset contains a mixture of single-hop and multi-hop questions, we only take the 3+ hops subset and name it BeerQA3+ for our evaluation.

**The specialized experts excel at their targeted reasoning types.** For example, the factual expert outperforms the dataset-specific few-shot baseline on NQ, TriviaQA, and SQuAD, and the math expert improves accuracy from 19.5 to 61.8 on GSM8K and from 41.5 to 92.2 on MultiArith. The only exception is that the factual expert is the best-performing model on HotpotQA—a multihop reasoning benchmark. This is because HotpotQA is also knowledge-intensive (Yang et al., 2018), and retrieval augmentation can be even more helpful than Chain-of-Thought reasoning.

**The specialized experts are worse on reasoning types outside of their expertise.** For instance, the factual expert underperforms the dataset-specific few-shot baseline on all math and commonsense datasets. Similarly, the math expert underperforms the baseline on all factoid QA datasets. This means that a single specialized QA model cannot generalize on the diverse types of questions and it motivates us to propose the MoRE system to combine the strengths of different experts in order to fare well on all types of reasoning questions.

### 4.3 MoRE Improves Generalizability

Here we focus on the full MoRE router that scores each expert's answer for answer selection. The second block in Table 1 compares MoRE with several other baselines:

- **Oracle Ensemble**: We compute the upper bound by taking the optimal answer for each question. Therefore, for each question, as long as one of the expert models got the correct answer, the accuracy will be 1.
- **Majority Vote**: We choose the most frequent answer string among the four expert models as the final prediction.
- **MaxProb**: We choose the answer with the highest confidence score.

MoRE with either the Codex answer selector or the random forest selector has better macro-average accuracy on the 12 datasets than any of the single-expert baselines (the first block in Table 1) and is also better than the majority vote or MaxProb baseline. In particular, MoRE with the random forest selector beats the best-performing expert (Commonsense Expert) by 8 points in macro-average accuracy and is significantly better than the Codex selector, demonstrating strong generalizability.

We emphasize that **we do not know the question type beforehand**. The single expert baselines do excel at their corresponding question types (e.g., factual expert performs the best on factual questions, even better than MoRE), but they perform terribly on other question types (e.g., the factual expert is much worse than normal dataset-specific few-shot prompting on math and commonsense questions). In contrast, for any given test question, our MoRE system's answer selector can select the best expert for that question without prior knowledge of its type. This selection process is crucial because there is no single expert model excelling across all types of questions. Therefore, it is this generalization accuracy (i.e., the "macro-average" accuracy column in Table 1) that we are highlighting as MoRE's core advantage, where MoRE scores 57.6 accuracy, outperforming all single-expert baselines in macro-average accuracy by large margins.

### 4.4 Question-Only Routing

In this section, we introduce the **Question-Only** setting, where we route based on the question alone. This means that we do not ask all four expert models for an answer; instead, we pick one expert and get the answer from it. Thus, we train the random forest router without any features that involve the expert predictions or their agreement. We also include two baselines for this setting: 1) randomly selecting an expert for each question; 2) a question-type oracle where we always route the question to the expert specialized in the corresponding question type (*e.g.*, we route all factual questions to the factual expert and all multi-hop questions to the multi-hop expert; which assumes knowledge of the question types).

This setup contrasts with the full MoRE router setting in Section 4.3, where all four expert models answer and then select the best answer. This requires four times more compute, but allows us to obtain more information for the expert selection.

The question-only routing approach beats single-expert baselines (Table 1, last block), but lags behind full MoRE. In particular, MoRE's question-only router has a macro-average accuracy of 50.2, slightly higher than the best single-expert (Commonsense Expert) with 49.6 accuracy, but significantly lower than MoRE with the full router (57.6 macro-average accuracy). In the remaining sections of the paper, we focus only on the MoRE

router given its strong performance, and study how to enable selective prediction.

## 5 MORE Improves Selective QA

The previous section has confirmed the generalizability strength of MORE, but it is still far from perfect. In fact, it is impossible for any QA system to be perfectly accurate on all questions, thus highlighting the importance of abstention—the system should not output an answer when it is likely to be wrong. For this goal, MORE has the important advantage of being more interpretable since users can understand how the system derives the final answer by inspecting each expert's prediction and the answer selection process. We demonstrate the benefits of such interpretability via evaluation on automatic abstention as well as human abstention.

### 5.1 Automatic Abstention

Traditionally, the decision to abstain or not is determined based solely on a confidence score. However, confidence scores of the generated answers can be poorly calibrated (Jiang et al., 2020; Si et al., 2022) for this purpose. A more effective approach is to train a calibrator to score the prediction's probability of being correct (Kamath et al., 2020; Ye and Durrett, 2021; Zhang et al., 2021). For MORE, we can easily use the answer selector as the calibrator to score the final predictions. Since MORE gathers predictions from multiple experts, it enables users to take advantage of the agreement among these expert systems as an additional useful signal apart from the confidence scores. To verify the effectiveness of such inter-expert agreement signals, we use the random forest selector from Section 3.2 to score model predictions, and ablate the impact of including inter-expert agreement features. We use the same MORE system with the random forest selector as the underlying QA system, which means that the QA accuracy would stay the same across all settings. We then compare the following three ways of scoring the final system predictions for automatic abstention:

- **MaxProb**: We directly take the selected answer's language modeling probability (as provided by the underlying Codex model) as the prediction's score.
- **RF Calibrator w/o Inter-Expert Agreement**: To tease apart the impact of inter-expert agreement features, we train the random for-

est classifier without any of the inter-expert agreement features described in Section 3.2.
- **MORE Calibrator**: We use the random forest classifier with all features in Section 3.2 as the calibrator. We simply take the classifier's predicted score on the selected answer as the score for the final prediction.

We use the following established metrics for evaluating the effectiveness of selective QA:

- **Area Under Curve (AUC)**: For any given threshold $\gamma$, there is an associated coverage and error rate (risk). We plot risk versus coverage and evaluate the area under this curve (AUC). This metric averages over all possible threshold $\gamma$, and lower AUC indicates better selective QA performance.
- **Coverage at Accuracy (Cov@Acc)**: We report the maximum possible coverage for a desired accuracy level. We report Cov@80% and Cov@90% in the table.
- **Effective Reliability (ER)**: Following Whitehead et al. (2022), we compute the score $\phi$ of each prediction as: (1) $\phi = 1$ if the system chooses to output an answer and the answer is correct (exact match equals 1); (2) $\phi = 0$ if the system chooses to abstain; (3) $\phi = -1$ is the system chooses to output an answer but the answer is wrong. The ER is then computed as the average of this score over the test set of size $n$: $\Phi = \frac{1}{n} \sum_x \phi(x)$. The threshold $\gamma$ for deciding whether to abstain or not is tuned on our dev set (which consists of 100 questions from each dataset) and applied on the test sets.

**Results** Our full MORE calibrator wins on on all metrics (Table 2) including AUC, Cov@80%, Cov@90%, and effective reliability. Interestingly, the random forest calibrator without the inter-expert agreement features is worse than the Max-Prob baseline (e.g., on AUC), which further highlights the benefit of having the inter-expert agreement as part of the calibrator design.

### 5.2 Human Abstention

We next verify that the expert-agreement and answer-selection information also help humans determine the correctness of the system's output.

**Setup** For the human study, we recruit 20 annotators from Prolific, who each annotated 20 randomly sampled questions. Our between-subject study has

| Method | AUC$_\downarrow$ | Cov@Acc=80%$_\uparrow$ | Cov@Acc=90%$_\uparrow$ | ER$_\uparrow$ |
|---|---|---|---|---|
| MaxProb | 34.8 | 32.4 | 12.4 | 17.5 |
| RF Calibrator w/o Agreement | 36.0 | 26.6 | 12.8 | 22.9 |
| MORE Calibrator | 28.3 | 45.9 | 34.3 | 33.4 |

Table 2: Incorporating inter-expert agreement features in the MORE calibrator improves selective QA as measured by all metrics and outperforms the MaxProb baseline by large margins. All results are the macro-average over 12 datasets.

| Condition | Decision Acc | ER | Accept Correct | Reject Wrong | Correct Conf | Wrong Conf | Time (Mins/20Qs) |
|---|---|---|---|---|---|---|---|
| Baseline | 57.0 | 9.5 | 75.0 | 36.8 | 0.69 | 0.59 | 15.0 |
| MORE | 67.5 | 19.5 | 89.4 | 43.8 | 0.78 | 0.67 | 13.2 |

Table 3: In human studies, 20 annotators (200 annotations) decide whether the system prediction is correct: 1) They achieve higher accuracy in deciding whether the final system output is correct when presented with information about the expert predictions and their scores (the MORE condition), which also corresponds to higher effective reliability (ER). 2) Showing expert information improves annotators' accuracy in both accepting correct answers and rejecting wrong predictions; 3) It boosts user confidence in both their correct and wrong judgments (although ideally we want the confidence on wrong judgments to be lower); 4) The MORE condition also takes less time for users to make decisions.

two conditions: (1) in the **baseline** condition, we present users with only the question and the final MORE prediction; (2) in the **MORE** condition, apart from the question and the final answer, we also present the predictions of each expert model along with the random forest classifier's scores of the candidate answers (interface in Appendix Figure 4). We also include a brief description of every expert's specialization in the task instruction to help annotators better understand the information. Half of the annotators were assigned to the baseline condition and the other half to the MORE condition. We provide an average compensation of $14.7 per hour and did not apply any additional screening apart from asking for proficient English speakers.

**Results**   For each question, we ask annotators to decide: (1) whether they think the final prediction is correct (binary judgment); and (2) what is their confidence in their own judgment on a scale of 1 to 5, which we will convert to a numerical value in range $[0, 1]$ for computing the average.

MORE improves both the accuracy and efficiency of human answer verification (Table 3). MORE improves annotators' accuracy of deciding whether the system prediction is correct from 57.0% to 67.5% ($p = 0.012$), which also corresponds to a jump in effective reliability from 9.5 to 19.5. When we break down the results into the accuracy of accepting correct model predictions

and rejecting wrong model predictions, the MORE condition improves accuracy in both categories. When measuring annotators' confidence in their judgment, their confidence increases in both correct and wrong judgment, as a result of seeing the additional inter-expert agreement information in the MORE condition.

Lastly and somewhat surprisingly, MORE's additional information did not slow people down: annotators spend an average of 13.2 minutes every 20 questions, compared to the average time of 15.0 minutes in the baseline condition, possibly because the lack of supporting evidence in the baseline condition makes the decision process difficult for people (similar effect as in Feng and Boyd-Graber (2022)). Interestingly, the automatic calibrator from MORE has an effective reliability score of 11.3 on the same sampled set annotators saw. This is higher than the human ER in the baseline condition (9.5) but lower than the human ER in the MORE condition, indicating that humans are able to capture additional cues that our automatic calibrator missed. Next, we examine those cases where humans effectively overruled MORE.

**Case Studies**   While humans largely rely on the expert selection and inter-expert agreement for abstention like the MORE calibrator, they sometimes also use background knowledge about the question (examples in Figure 2). In the first example from HotpotQA, the annotator trusted the wrong

**Question [HotpotQA]**: Which documentary was filmed first, *Almost Sunrise* or *Hail! Hail! Rock 'n' Roll*?

**Expert Predictions**:
- 🌐 Factual: *Almost Sunrise* (0.45)
- 🕸 Multihop: *Hail! Hail! Rock 'n' Roll* (0.61)
- 🔢 Math: *Almost Sunrise* (0.35)
- 🧑‍🤝‍🧑 Commonsense: *Almost Sunrise* (0.38)

**Final Selected Prediction**: *Almost Sunrise*
**Gold Answer**: *Hail! Hail! Rock 'n' Roll*
**MoRE**: **Abstained**
**Human**: **Trusted the wrong prediction**
**Justification**: Don't expect multihop to get simple answers right, all others answer the same.

**Question [QASC]**: Where is the main place where food absorption occurs?

**Expert Predictions**:
- 🌐 Factual: *microscopic vessels* (0.30)
- 🕸 Multihop: *small intestines* (0.67)
- 🔢 Math: *The small intestines is the main place where food absorption occurs.* (0.03)
- 🧑‍🤝‍🧑 Commonsense: *small intestines* (0.64)

**Final Selected Prediction**: *small intestines*
**Gold Answer**: *small intestines*
**MoRE**: **Trusted the correct prediction**
**Human**: **Trusted the correct prediction**
**Justification**: The factual expert should have gotten this right but all other 3 answers are the same with high scores.

**Question [MuSiQue]**: When was the celestial body that all objects focus on according to Kepler's laws discovered to be the center of the solar system?

**Expert Predictions**:
🌐 Factual: *1610* (0.15)   🕸 Multihop: *17th century* (0.54)
🔢 Math: *1687* (0.04)   🧑‍🤝‍🧑 Commonsense: *17th century* (0.28)

**Final Selected Prediction**: *17th century*   **Gold Answer**: *as early as the 3rd century BC*
**MoRE**: **Trusted the wrong prediction**   **Human**: **Abstained**
**Justification**: I think it should be much further back. The AI probably key'd on Kepler instead of the discovery of the sun being the Center of the Solar system.

Figure 2: Three examples of MoRE automatic abstention and human abstention. For each example, we show the question, each reasoning expert's prediction along with its score, the best prediction selected by the random forest classifier, the actual gold answer, the abstention decision by MoRE and human annotators as well as the annotators' justification. **Humans often rely on inter-expert agreement and their own understanding of how these expert models work.**

prediction because three of the expert models made the same prediction and the annotator didn't recognize that the question is multihop (in fact the multihop gave the correct answer but it's not selected as the final prediction). In the second example from QASC, although the annotator judged the question to be a factoid question, they went with the consensus of the other three expert models. These two examples show that humans rely on both the match between the question type and corresponding expert strength, as well as the inter-expert agreement for their judgment. In the third example from MuSiQue, the annotator inferred why the model made the particular prediction and successfully spotted the mistake. Such external knowledge may partially account for why humans get better abstention effective reliability than MoRE.

## 6 Related Work

**Specialized Prompting and Prompt Ensemble** To better elicit knowledge and reasoning from LLMs, many prompting methods have been proposed, such as Least-to-Most (Zhou et al., 2023) and Self-Ask Prompting (Press et al., 2022) for multi-step reasoning, and Program-of-Thought (Chen et al., 2022) and Declarative

Prompting (Ye et al., 2023) for symbolic reasoning. Unlike these works, our goal is to combine the strengths of all the specialized language models empowered with these specialized prompting techniques for better generalizability and selective QA. Another line of work ensembles multiple answers from LLMs: Wang et al. (2023) samples multiple answers with a high temperature during decoding and selects the final answer by majority vote; while Li et al. (2022b) constructs different prompts by selecting different demonstration examples and trains a verifier to perform weighted voting on the answers. Unlike these approaches, we create reasoning experts with different specializations in order to achieve generalizability and leverage the inter-expert agreement features for both answer selection and abstention.

**Modular LM and Mixture-of-Experts** One classic example towards modular language models is Mixture-of-Experts (Jacobs et al., 1991), which is adopted in scaling sparse Transformer models like GShard (Lepikhin et al., 2020), Switch-Transformer (Fedus et al., 2022), BASE-Layer (Lewis et al., 2021), DEMIX (Gururangan et al., 2021), Branch-Train-Merge (Li et al., 2022a),

and C-BTM (Gururangan et al., 2023). Unlike these Mixture-of-Experts, our MORE system does not route at the token level but rather designs specialized experts and routes the entire question to the best expert. The most similar works to ours are Puerto et al. (2023) and Jiang et al. (2023), where each expert model generates an entire response to the query and a reranker then selects the best answer. However, unlike all these prior works, each specialized model in MORE is carefully designed to excel in a particular reasoning type (rather than domain experts like most prior works), allowing for better complementary strengths across reasoning types, and to the best of our knowledge, we are the first study to focus on ensembling experts under the more practical selective QA setting.

**Generalizable QA and Multitask Learning**
MRQA (Fisch et al., 2019) benchmarked the domain generalizability of machine reading comprehension models and similar to Talmor and Berant (2019): QA models trained on one domain often fail to generalize on others. To improve generalizability, Khashabi et al. (2020) trained a unified model on a large collection of QA datasets, while Friedman et al. (2021) trained lightweight adapters for domain generalization. Unlike these works, we focus on the more challenging setting of generalizing across different reasoning types, and we take a different approach by ensembling multiple specialized models. Beyond QA, a growing line of work trains multitask models via multitask training (Zhong et al., 2021; Min et al., 2022) or instruction tuning (Mishra et al., 2021; Wei et al., 2022a; Wang et al., 2022), which allows LLMs to extrapolate across different types of tasks. However, such fine-tuned models (with multitask or instruction tuning) still suffer from poor interpretability, while our proposed framework allows users to inspect the internal expert selection process for better interpretability.

**Selective Prediction**    Several prior works studied training effective calibrators to decide when to abstain. Kamath et al. (2020) studied selective QA under domain shifts where they showed that training a random forest calibrator is better than relying on LM probability alone. Ye and Durrett (2021) additionally included local explanation features to improve the calibrator, and Zhang et al. (2021) embedded questions as dense vector features to improve the calibrator. Xie et al. (2022) focused

specifically on multihop questions and achieved benefits from incorporating question decomposition information in the calibrator. Garg and Moschitti (2021) filtered unanswerable questions based on model confidence to improve computation efficiency. Rodriguez et al. (2019) studied incremental question answering (Quizbowl) where calibration is an intrinsic part of the task in order to decide the best timing for making a prediction ("buzzing"). Our work contributes to this line of work by showing the benefit of designing a more interpretable QA system where the inter-expert agreement features are helpful for calibration and selective QA.

## 7    Conclusion

We proposed the MORE framework where we construct a pool of specialized QA models that excel at different reasoning types, and then train an answer selector to select the best answer among them. Experiments on 12 datasets covering four reasoning types demonstrate that MORE achieve better generalizability than all baselines. More importantly, the inter-expert agreement features in MORE offer useful signals for training effective calibrators that improve selective QA and also improve human verification of the system's final predictions.

While we focused on prompting LLMs as specialized experts, the idea of combining the strengths of diverse experts can extend to any type of specialized models, even non-neural ones such as traditional information retrieval models, which is an interesting avenue for future work. Additionally, future work could also explore other possible explanations to facilitate users' calibration and abstention, such as better explaining the strengths and weaknesses of individual specialized expert models. Such efforts are especially important for high-stakes settings that require careful fact-checking or verification of the system outputs (Si et al., 2023b).

## Limitations

**Model Coverage**    We only focused on the Codex model for the experiments due to its strong performance on QA tasks (at the time of writing this paper). It would be interesting to verify our framework on different LLMs, especially open-source models. Moreover, future work could move beyond using prompted LLMs as the specialized experts and instead ensemble more heterogeneous expert models such as models finetuned on particular reasoning types or non-Transformer models.

**Reasoning Type Coverage** We experimented with four representative reasoning types but there exist many more question types that could possibly occur in real-life applications, such as questions with multiple answers, ambiguous questions, and questions with false presuppositions. It would be interesting for future work to study how to extend our framework to also tackle these additional reasoning types, for example by designing and adding new specialized models.

**Beyond QA** While we only focused on QA evaluation, another interesting direction for future work is to extend our idea beyond just QA, for example for general-purpose language modeling. This likely requires re-designing the evaluation pipeline and implementing specialized expert models that are not only performant for QA tasks but for language generation in general.

## Ethical Considerations

**Human Study** Our human study has been exempted by the Institutional Review Boards, and we compensate annotators an average of $14.7 per hour, well above the minimum wage in the US. We do not expect any harm during the entire annotation process.

**Broader Impact** Our work improves the reliability of QA systems in the wild and improves the long-standing problem of users over-trusting answers from black-box AI systems. We believe that our interpretable MORE system can inspire more future work on designing AI systems where humans can verify the answers and calibrate their trust appropriately in order to avoid being misled by erroneous AI outputs.

## Acknowledgement

We thank Ruiqi Zhong, Tianyu Gao, Xi Ye, and Daniel Khashabi for their helpful discussion. We thank Peng Qi for providing the BeerQA evaluation data. We thank Navita Goyal for providing helpful advice on building the human study interface. Chenglei Si completed this work back when he was an undergraduate researcher at UMD and he thanks all members of the UMD CLIP lab for their support throughout his undergraduate research journey. Chen Zhao is supported by Shanghai Frontiers Science Center of Artificial Intelligence and Deep Learning, NYU Shanghai. This work is also supported by Meta AI through Dynabench Data Collection and Benchmarking Platform.

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

# A  Appendix

## A.1  Prompts for Reasoning Experts

Figure 3 shows the actual prompt design for the four specialized reasoning experts in MORE.

## A.2  Interface for Human Study

Figure 4 shows the annotation interface for our human abstention study. We provide instructions for the task, describe the reasoning experts' strengths, then show the test questions with all expert predictions and scores. We then ask annotators to determine the correctness of the final prediction, their confidence, as well as their justification. In the baseline condition, the expert prediction panel is omitted.

## A.3  Features for Training the Classifier

Below we list all the features used to train our random forest classifier that scores expert predictions.

- **Specialized Expert Type**: a one-hot four-dimensional vector.
- **Question Characteristics**: question word, question length, and the number of numerical values in the question.
- **Answer Characteristics**: the probability of the generated output (multiplying each token's likelihood and normalizing by length as in Si et al. (2023a)), the length of the generated answer, the overlap between the question and the predicted answer, the number of numerical values in the answer, overlap between the answer and retrieved or generated passages, length of CoT rationales, overlap between questions and rationales, overlap between answers and rationales, the number of times the answer appears in the rationale, and the number of numerical values in the rationale.
- **Factual and Commonsense Experts' Contexts**: the number of numerical values in the retrieved or generated passages, the number of overlapping tokens between questions and passages, and the passage length.
- **Inter-Expert Agreement**: the frequency of the predicted answer among all four experts' predictions, token overlap among the experts' outputs.

Some of these features are expanded upon prior works on selected QA (Rodriguez et al., 2019; Ye and Durrett, 2021; Zhang et al., 2021).

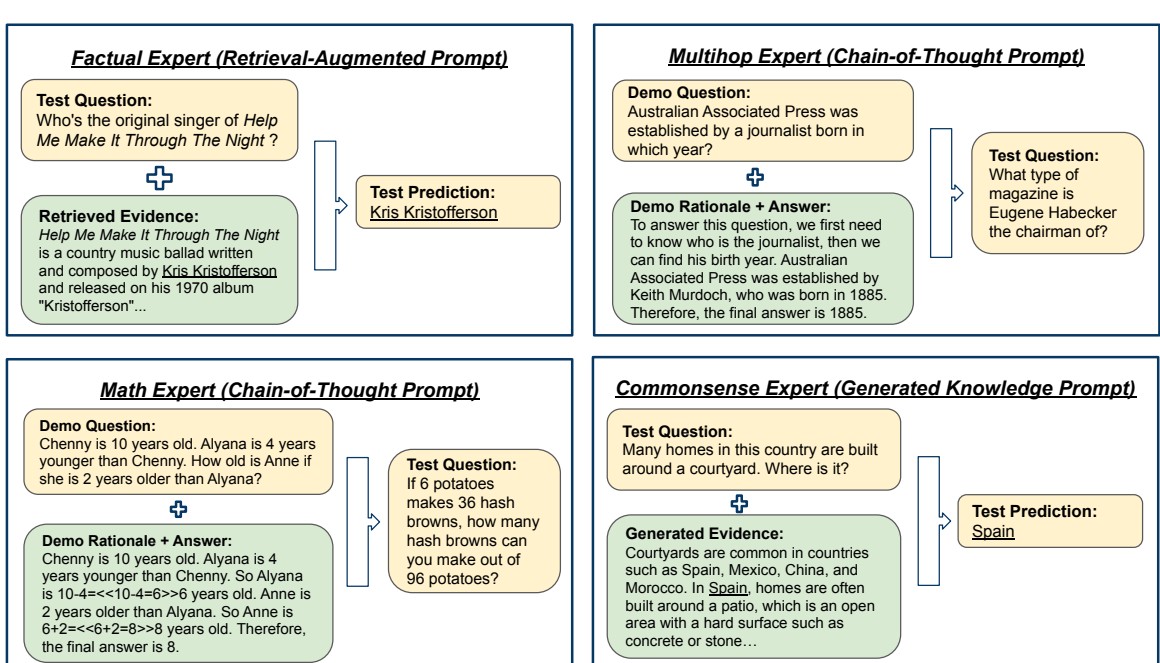

Figure 3: The four specialized QA models in MoRE, implemented by applying specialized prompts on Codex. For the factual expert, the demo examples are randomly sampled examples from NQ and we append retrieved evidence from Wikipedia for each test question; for the multihop expert, we use question and rationale-answer pairs from HotpotQA as the prompt; for the math expert, we use question and rationale-answer pairs from GSM8K as the prompt; for commonsense expert, we use random examples from CommonsenseQA as the prompt, and we use the same LLM to generate related background knowledge to append to each test question.

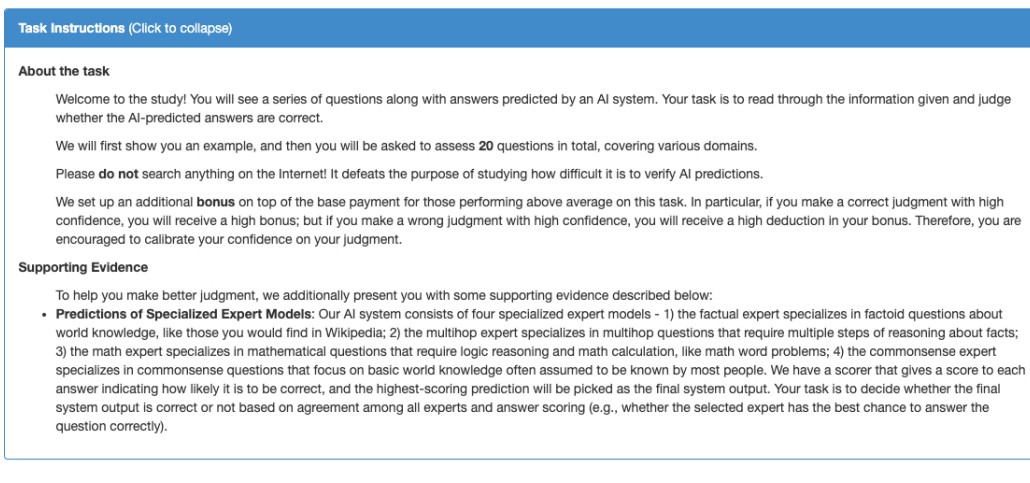

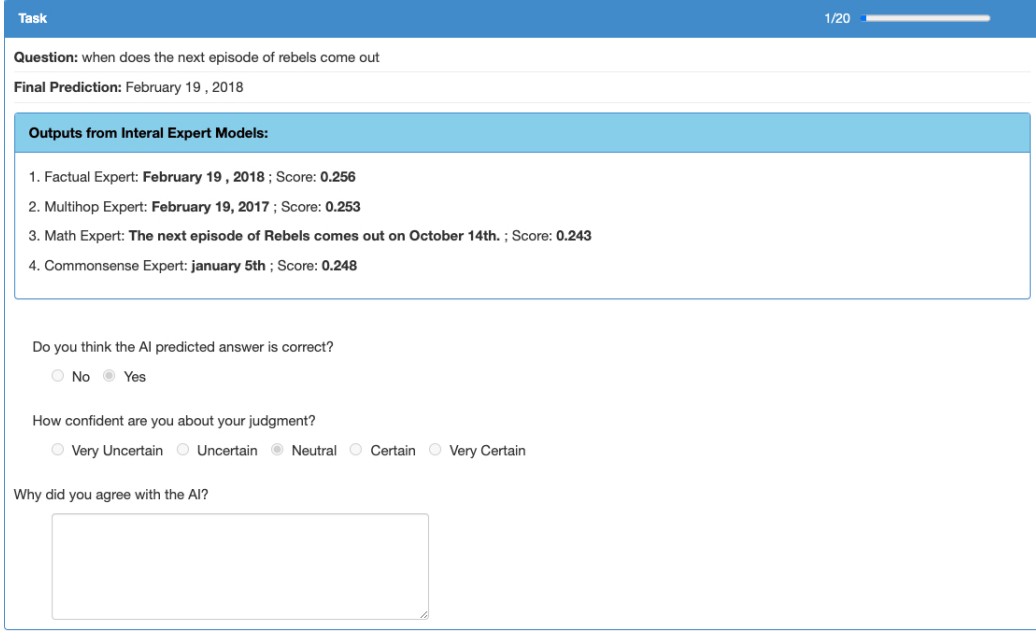

Figure 4: Our annotation interface for the human abstention study.