# OpenReview forum: "Getting MoRE out of Mixture of Language Model Reasoning Experts"
_EMNLP/2023/Conference — EMNLP 2023 Findings_

### Official Review · Reviewer_MmuJ · 2023-08-04

**Soundness:** 4

**Excitement:**

3: Ambivalent: It has merits (e.g., it reports state-of-the-art results, the idea is nice), but there are key weaknesses (e.g., it describes incremental work), and it can significantly benefit from another round of revision. However, I won't object to accepting it if my co-reviewers champion it.

**Paper Topic And Main Contributions:**

This paper proposed a Mixture-of-Reasoning-Experts framework that ensembles diverse specialized language models by using different prompting skills to build factual, multihop, mathematical, and commonsense reasoning models respectively, and designing an answer selector to select the best answer or abstain from answering.

This framework aims at generalizing, answering selectively, and enhancing interpretability. It also outperforms any single specialized model across all four diverse reasoning types.

**Questions For The Authors:**

Can the authors share the results of other different classifiers mentioned in lines 235-239?

**Reasons To Accept:**

Thorough evaluation across twelve datasets and detailed ablation studies.

The proposed method can serve as an inspiration and be generalized further to other tasks by implementing specialized expert models and especially redesigning the evaluation pipeline.

**Reasons To Reject:**

QA type coverage is not representative. The authors fail to give a proper explanation of the choices of QA types. Also, Factual reasoning and multihop reasoning often have some overlaps.

The experiments are only conducted with Codex and random forest. More diverse LLMs should be tested to further prove the effectiveness of the proposed method. The authors should also present the results of different classifiers such as boosting algorithms and BERT to show random forest is the best fit. (see the question)

**Reproducibility:**

4: Could mostly reproduce the results, but there may be some variation because of sample variance or minor variations in their interpretation of the protocol or method.

**Reviewer Confidence:**

4: Quite sure. I tried to check the important points carefully. It's unlikely, though conceivable, that I missed something that should affect my ratings.

**Typos Grammar Style And Presentation Improvements:**

The result of Appendix A.1 and Table 4 in the appendix should be moved back to the main paper since it is a main experiment to justify the importance of an answer selector, which is the major contribution of this paper.

---

> ### Author Rebuttal · Authors · 2023-08-27
>
> Thank you for the thoughtful review! To address your concerns:
>
> 1) Re “QA type coverage is not representative. The authors fail to give a proper explanation of the choices of QA types.”:
>
> Our list of QA types is based on existing QA literature. E.g., in “QA Dataset Explosion: A Taxonomy of NLP Resources for Question Answering and Reading Comprehension” by Rogers et al., QA skills are split into 1) Retrieval (corresponding to our Factual questions and Commonsense questions); 2) Multi-step (corresponding to our Multihop questions); 3) Numeric skills (corresponding to our Maths questions) and several other sub-categories.
>
> It is definitely true that there are some other categories of questions that we didn’t have the bandwidth to cover (e.g., long-form QA, analogy questions, subjective questions, etc.), which generally have fewer available datasets to benchmark on. We acknowledge this is a limitation of our work and will discuss it in detail in the limitations section of the paper. However, in principle, there is nothing stopping our MoRE framework from extending to support all these question types as long as there are data resources available.
>
>
> 2) Re “The authors should also present the results of different classifiers such as boosting algorithms and BERT to show random forest is the best fit.” “Can the authors share the results of other different classifiers mentioned in lines 235-239?”:
>
> Yes, we have tried these different classifiers. E.g., for macro-average accuracy across all datasets, MoRE with a finetuned BERT selector gets 56.4, an AdaBoost selector gets 56.3, and an SVM selector gets 56.0, all lower than the random forest selector (57.6). We will include the table for all these alternative methods we’ve tried along with exact implementation details.
>
> 3) Re “The result of Appendix A.1 and Table 4 in the appendix should be moved back to the main paper since it is a main experiment to justify the importance of an answer selector, which is the major contribution of this paper.”:
>
> We will do it for the next version, thanks for pointing it out!

---

### Official Review · Reviewer_G4bF · 2023-08-06

**Soundness:** 3

**Excitement:**

3: Ambivalent: It has merits (e.g., it reports state-of-the-art results, the idea is nice), but there are key weaknesses (e.g., it describes incremental work), and it can significantly benefit from another round of revision. However, I won't object to accepting it if my co-reviewers champion it.

**Paper Topic And Main Contributions:**

This paper works on question answering (QA) using LLMs. The main aim is to generalize a single LLM across different QA tasks.  To achieve this goal, this paper proposes a  mixture-of-reasoning-experts (MoRE) method to exploit factual, multihop, mathematical and commonsense reasoning experts.  Experimental results on 12 QA tasks and human study show that the proposed method is both more effective and interpretable than the investigated baselines.

**Questions For The Authors:**

1. In Table 1,   more specific information about the MORE - Codex Selector setting should be provided.
2. How to generate the confidence score of each prediction from a different expert?


**Reasons To Accept:**

1. The motivation for generalizable QA is well-positioned and the proposed idea is reasonable.
2. The code and data are released.
3. The macro-average performances across 12 datasets are significantly improved compared to multiple baselines.
4. The analysis section contains human study to show the usefulness of the selective process.


**Reasons To Reject:**

1. Although the main focus of MoRE is for reasoning type generalization,  the performances of mixture experts are not better than the single expert model on factual and multihop tasks.  The overall method adds complexity compared to single expert models.  These together result in less attractive contributions.
2. In table 1,   the main improvements of MoRE are most from the 3 Math QA tasks and 2 common sense tasks.   WIthout the math tasks, the improvements are less impressive.  This observation weakens the claim of reasoning type generalization of the proposed method.  The relative benefits from different experts are not investigated.  Which expert benefit the math task the most and why?   Why factual, multihop and commonsense tasks can greatly improve the performance of math tasks?  I cannot see that this is very straightforward.
3. With the outcome from multiple experts,  can we just directly feed the output of the multiple experts and let LLMs (say ChatGPT) choose or generate/revise the correct answer?


**Reproducibility:**

2: Would be hard pressed to reproduce the results. The contribution depends on data that are simply not available outside the author's institution or consortium; not enough details are provided.

**Reviewer Confidence:**

4: Quite sure. I tried to check the important points carefully. It's unlikely, though conceivable, that I missed something that should affect my ratings.

---

> ### Author Rebuttal · Authors · 2023-08-27
>
> Thank you for the thoughtful review! To address your concerns:
>
> 1) Explanation for “the performances of mixture experts are not better than the single expert model on factual and multihop tasks”:
>
> We want to clarify how to interpret the results of these single expert models. Comparing MoRE with these individual specialized experts on their corresponding specialized reasoning types is actually an unfair comparison, and rather it is the generalization accuracy (“macro-average” column in Table 1) we are trying to compare and demonstrate MoRE’s advantage, where MoRE scores 57.6 accuracy as compared to 38.4 of the factual expert and 47.2 of the multihop expert (in fact MoRE outperforms all baselines in macro-average accuracy by large margins).
>
>
> Using the factual expert on factual questions is like an “oracle” where you already know the questions will be knowledge-intensive factual questions, but in real-life applications, such an assumption is impossible because the questions can be from different categories and we want our system to generalize. In fact, if we only use the factual expert to answer all questions, it will suffer terribly on other question types beyond factual questions (e.g., on math and commonsense questions) and result in a low macro-average across all datasets (38.4). The same applies to the multihop expert which does well on multihop questions but very poorly on factual or commonsense questions, resulting in a low macro-average as well (47.2). In contrast, our proposed MoRE system can select the appropriate expert for each test question to combine the strengths of these different experts, resulting in the best macro-average across all 12 datasets and 4 reasoning types (57.6), outperforming all other baselines in terms of the generalization capability. In fact, reviewer azTT raised a similar question and we refer you to the second point of our response to reviewer azTT if you are interested to see more discussion on this.
>
>
> 2) Regarding the comment “The overall method adds complexity compared to single expert models.”:
>
> The MoRE system only requires training a simple random forest classifier as the expert selector in order to combine the strengths of existing specialized LLMs, which takes minimal time or compute resources. So we believe it is actually our method’s advantage that it is simple and allows greater generalizability and interpretability than single expert models.
>
>
> 3) Explanation for “the main improvements of MoRE are most from the 3 Math QA tasks and 2 common sense tasks; without the math tasks, the improvements are less impressive.”:
>
> Although it is true that the specialized math and multihop experts perform particularly well on the math and multihop questions (large gains compared to the dataset-specific few-shot baseline), the main advantage of MoRE lies in its ability to generalize across all question types and get the best macro-average. For example, if you always use the math expert for all question types, you will indeed do very well on math questions, but the math expert will perform terribly on factual, multihop, and commonsense questions, resulting in a low macro-average (46.2) compared to MoRE (57.6). So, it’s not true that MoRE only works well because the math expert and multihop expert are particularly effective (always using the math expert or always using the commonsense expert is much worse than MoRE in terms of macro-average), but rather, it is the expert selection and ensembling that truly enable MoRE’s superior generalization accuracy, which is a major contribution of the paper.
>
> The math expert and commonsense expert work particularly well on their corresponding question types as you mentioned, which is mainly because the specialized math chain-of-thought and generated knowledge prompting methods are very effective towards these questions. Factual questions and multihop questions benefit less from the specialized prompting methods since these questions tend to be harder (e.g., requiring significant world knowledge). However, these specialized experts can be further improved in the future with the rapid development of more powerful base LLMs and better specialization methods (either through prompting or specialized tuning), and our framework will still be applicable and effectively combine the strengths of the different specialized expert models.
>
> The overall gain may be even larger if the individual expert gets better on their corresponding reasoning types, but we also highlight that our main conclusion on generalization gains of MoRE would still hold no matter the scale of these individual expert gains. This is because individual specialized experts cannot generalize well, so their macro-average accuracy will be low across all datasets, but MoRE can combine the strengths of all experts for better generalization.
>
>
> 3) Re “The relative benefits from different experts are not investigated. Which expert benefits the math task the most and why? ”:
>
> This is a great point! We agree it will be insightful to analyze how much each expert model contributes on each dataset. We will add a detailed breakdown of each expert’s contribution to the overall MoRE system on each dataset in the next version of the paper. But in summary, the specialized expert usually contributes the most to the corresponding question type. For example, on math questions, the expert selector would select the math expert’s answer most of the time (70% of the time), leading to a high accuracy on math questions; but there are also some cases where the selector chooses other experts (e.g., the multihop expert is selected 20% of the time) for math questions, which is reasonable since some math questions also require multihop reasoning.
>
> 4) Re “Why factual, multihop, and commonsense tasks can greatly improve the performance of math tasks?”:
>
> It is true that sometimes these expert models can benefit question types beyond their target type. For example, the multihop expert improves performance on math questions since many math questions do require multistep reasoning (e.g., from GSM8K: “Julie is reading a 120-page book. Yesterday, she was able to read 12 pages and today, she read twice as many pages as yesterday. If she wants to read half of the remaining pages tomorrow, how many pages should she read?”). Similarly, the commonsense expert also shows some improvement on math questions, because the generated knowledge prompting actually generates useful sub-steps for solving the math problems (e.g., the commonsense expert would generate the sub-steps “Colorado City uses 40% of the water from the Colorado River. If 80% of that water is used for industrial purposes, then 20% is used for non-industrial purposes. Since Colorado City uses 40% of the river's total water, 20% of that is used for non-industrial purposes. Therefore, the city uses 8% (20% of 40%) of the river's total water for non-industrial purposes.” to help solve the math question: “Colorado City uses 40% of the water from the Colorado River. If 80% of that water is used for industrial purposes, what percent of the river's total water is used by the city for non-industrial purposes?”).
>
> The factual expert does not help on math tasks (in fact it performs even worse than dataset-specific few-shot prompting), but it does improve on some multihop QA datasets like HotpotQA, since many of these questions also require external factual knowledge (e.g., from HotpotQA: “Who is the director of the 2003 film which has scenes in it filmed at the Quality Cafe in Los Angeles?”).
>
> 5) Re “can we just directly feed the output of the multiple experts and let LLMs (say ChatGPT) choose or generate/revise the correct answer”:
>
> Yes, that’s exactly the few-shot Codex selector described on lines 240- 250. This method gets a macro-average accuracy of 51.4 as shown in Figure 1, worse than MoRE with the random forest selector (57.6). We will add more details for its implementation in the main paper as you suggested.
>
>
> 6) Re “How to generate the confidence score of each prediction from a different expert?”:
>
> We multiply each token’s probability and then normalize by the length (total number of tokens) of the generated answers. E.g, suppose the log token probs are stored in a list token_probs, the confidence of the answer is computed as confidence =  np.exp((np.mean(token_probs))). We will include all these details in the next version of the draft to make it clearer.

---

### Official Review · Reviewer_azTT · 2023-08-07

**Soundness:** 4

**Excitement:**

3: Ambivalent: It has merits (e.g., it reports state-of-the-art results, the idea is nice), but there are key weaknesses (e.g., it describes incremental work), and it can significantly benefit from another round of revision. However, I won't object to accepting it if my co-reviewers champion it.

**Missing References:**

https://arxiv.org/abs/2207.05221
https://aclanthology.org/2022.repl4nlp-1.23/
https://arxiv.org/pdf/2109.07009.pdf

Also few references in the 'Reasons to Reject' section

**Paper Topic And Main Contributions:**

This paper is on the interesting area of single model generalizing across question types that require distinct reasoning abilities. Authors provide empirical evidence that state-of-the-art LLMs suffer from poor generalizability on reasoning types
beyond those seen in the prompt. They propose a Mixture-of-Reasoning Experts (MORE) framework combining specialized language models. The main idea to leverage agreement among the specialized experts to select the best answer for each question, or to abstain from answering. They evaluate the efficacy of this method across 12 datasets.

**Questions For The Authors:**

See above

**Reasons To Accept:**

- interesting and timely problem statement.
- well-written paper.
- mixture of expert is an interesting solution to this problem. Bringing in selective answering angle to this is super-interesting. Note that, hallucination/trustworthiness is a super important problem.
- human study targeting human users that interpret model output is useful.

**Reasons To Reject:**

I need answer to these questions. These are not necessarily 'reasons to reject'.
- Authors mention UnifiedQA, but I did not see a comparison with the multitasking baseline in Table 1. Mixture of experts in m (m=4 in this paper) models for n tasks, multitasking model is 1 model for n tasks. There might be positive inter-task influence in multitask learning that could improve the final task performance. Wondering if authors tried this already?

- The recent instruction tuning framework (https://aclanthology.org/2022.acl-long.244.pdf, https://openreview.net/pdf?id=gEZrGCozdqR) has improved the multitask learning framework and provided added inter-task inductive bias for generalization. Wondering if authors have experimented with instructions in multitask learning or as part of their mixture of experts modelling?

- Table 1, some cases like the factual case, the proposed ensemble expert is worse than the single expert, whereas the performance improvement is higher in mathematical reasoning cases. Any explanation on this behavior? Does this mean: better selection of reasoning types (say replacing factual category by something else) will improve the efficacy of MORE further?

**Reproducibility:**

3: Could reproduce the results with some difficulty. The settings of parameters are underspecified or subjectively determined; the training/evaluation data are not widely available.

**Reviewer Confidence:**

4: Quite sure. I tried to check the important points carefully. It's unlikely, though conceivable, that I missed something that should affect my ratings.

---

> ### Author Rebuttal · Authors · 2023-08-27
>
> Thank you for the thoughtful review! To answer your questions:
>
> 1) Reasons for not comparing with the multitasking and instruction tuning baselines:
>
> We agree it would be interesting to try multitask finetuning or instruction tuning on the backbone LLM (Codex) and see how well it generalizes. We did not include it as a baseline because: 1) We emphasize that interpretability for users is a crucial goal of this work because we want users to be able to calibrate their confidence and avoid trusting wrong predictions which may cause serious trouble in real applications. Finetuned models (with multitask or instruction tuning) would still suffer from poor interpretability since it is still one single black-box LLM; while our proposed ensembling framework allows users to inspect the internal expert selection process for better interpretability.  2)  Intuitively, it would be difficult to distill all these different reasoning abilities into one single model, e.g., for knowledge-intensive factual questions, our factual expert with retrieval augmentation can retrieve relevant knowledge that the pretrained LLM may not know, but simply tuning on factual QA data may not unlock this ability in the LLM. (And note that our MoRE system also handles the problem of determining when to use this retrieval-enhanced factual expert.)  3) Unfortunately, finetuning Codex is still inaccessible from the OpenAI API that we are using (and other LLMs that allow finetuning generally perform much worse on reasoning tasks so we didn’t use them).
>
> Thank you for providing missing references to important related work - the lines of work on instruction tuning, selective prediction, and calibration are all very relevant and inspiring. We will add all of them to the next version and explicitly acknowledge the missing comparison with multitask training and instruction tuning.
>
> 2) Explanation for “some cases like the factual case, the proposed ensemble expert is worse than the single expert, whereas the performance improvement is higher in mathematical reasoning cases”:
>
> We want to highlight that the setting we are trying to target is to generalize across question types **without** knowing the question types beforehand. The single expert baselines do excel at their corresponding question types (e.g., factual expert performs the best on factual questions, even better than MoRE), but they perform terribly on other question types (e.g., the factual expert is much worse than normal dataset-specific few-shot prompting on math and commonsense questions).
>
> In contrast, for any given test question, our MoRE system’s answer selector can select the best expert for that question without prior knowledge of its type. This selection process is crucial because there is no single expert model excelling across all types of questions. Therefore, it is this generalization accuracy (i.e., the “macro-average” accuracy column in Table 1) that we are highlighting as MoRE’s core advantage, where MoRE scores 57.6 accuracy as compared to 38.4 of the factual expert (in fact MoRE outperforms all baselines in macro-average accuracy by large margins).
>
> We see a larger improvement in math reasoning as compared to factual questions because: 1) The math expert works particularly well on math questions as compared to standard few-shot prompting; 2) For the answer selector of MoRE, it is easier to identify math questions due to features like the presence of numbers.
>
> This connects to your question - whether “better selection of reasoning types (say replacing factual category by something else) will improve the efficacy of MoRE further”. It is true that the overall gain may be larger if the individual expert gets even better on their corresponding reasoning types, but we highlight that our main conclusion on generalization gains of MoRE would still hold no matter the scales of these individual expert gains. This is because individual specialized experts cannot generalize well, so their macro-average accuracy will be low across all datasets, but MoRE can combine the strengths of all experts for better generalization.
>
> To summarize: Comparing the MoRE system accuracy with the single-expert baselines on their corresponding question types is an unfair comparison since in reality we do not know the question type in advance and have to rely on the answer selector (if you use the factual expert model all the time you will just suffer on other question types and get a macro-average of only 38.4 like Table 1 shows). In the realistic setting of not knowing the question types, our proposed MoRE system is much better than any baseline in terms of macro-average (57.6)  across all datasets. We will revise the paper to clearly highlight the interpretation of these single-expert baseline results in comparison with the MoRE results.

---

### Meta-Review · Area_Chair_MDRJ · 2023-09-11

**Recommendation:** 3

**Metareview:**

*Summary of the work:* The work explores a mixture (ensemble) of "experts" to create more generalizable models. While gaining generalizability, the work also aims for a more interpretable aggregation of experts' results.

*Summary of reviewer comments:*
Based on my reading, most of the comments by the reviewers involve the expansion of the experiments. For example, azTT suggested comparison with multi-tasking-based baselines (which are not particularly interpretable). Others ask for more analysis on the contribution of the experts (G4bF) and analysis using more models (MmuJ). Implicit to these comments (including those by azTT) is the nature (pros/cons) of the suggested "interpretability" (which leads to "selective QA" which may be attainable with other forms of ensembling). After all, the interpretability here does not allow one to find out "why" a certain "expert" was chosen over the other ones.

In summary, the fact that the reviewers are asking for more experiments is not a killer issue. However, it is an indication that they expected a more rigorous set of analyses.


*Overall recommendation:*
Overall, it is a worthwhile study in a promising direction, though based on the concerns raised in the reviews, it is not clear a "accept." This is evident in that all the reviewers have indicated an "ambivalent" level of excitement for this work. As such, I will recommend this work for the "findings" track.

---

### Decision · Program_Chairs · 2023-10-07

**Decision:**

Accept-Findings

**Comment:**

*Summary of the work:* The work explores a mixture (ensemble) of "experts" to create more generalizable models. While gaining generalizability, the work also aims for a more interpretable aggregation of experts' results.

*Summary of reviewer comments:*
Based on my reading, most of the comments by the reviewers involve the expansion of the experiments. For example, azTT suggested comparison with multi-tasking-based baselines (which are not particularly interpretable). Others ask for more analysis on the contribution of the experts (G4bF) and analysis using more models (MmuJ). Implicit to these comments (including those by azTT) is the nature (pros/cons) of the suggested "interpretability" (which leads to "selective QA" which may be attainable with other forms of ensembling). After all, the interpretability here does not allow one to find out "why" a certain "expert" was chosen over the other ones.

In summary, the fact that the reviewers are asking for more experiments is not a killer issue. However, it is an indication that they expected a more rigorous set of analyses.


*Overall recommendation:*
Overall, it is a worthwhile study in a promising direction, though based on the concerns raised in the reviews, it is not clear a "accept." This is evident in that all the reviewers have indicated an "ambivalent" level of excitement for this work. As such, I will recommend this work for the "findings" track.